# Gallop Racing Shifts Mature mRNA towards Introns: Does Exercise-Induced Stress Enhance Genome Plasticity?

**DOI:** 10.3390/genes11040410

**Published:** 2020-04-09

**Authors:** Katia Cappelli, Samanta Mecocci, Silvia Gioiosa, Andrea Giontella, Maurizio Silvestrelli, Raffaele Cherchi, Alessio Valentini, Giovanni Chillemi, Stefano Capomaccio

**Affiliations:** 1Dipartimento di Medicina Veterinaria, University of Perugia, 06126 Perugia, Italy; katia.cappelli@unipg.it (K.C.); andrea.giontella@unipg.it (A.G.); Maurizio.silvestrelli@unipg.it (M.S.); stefano.capomaccio@unipg.it (S.C.); 2Centro di Ricerca sul Cavallo Sportivo, University of Perugia, 06126 Perugia, Italy; 3SCAI-Super Computing Applications and Innovation Department, CINECA, 00185 Rome, Italy; s.gioiosa@cineca.it; 4AGRIS, Servizio Ricerca Qualità e Valorizzazione delle Produzioni Equine, Ozieri, 09127 Sassari, Italy; rcherchi@agrisricerca.it; 5Dipartimento per l’Innovazione Nei Sistemi Biologici, Agroalimentari e Forestali, Università della Tuscia, 01100 Tuscia, Italy; alessio@unitus.it (A.V.); gchillemi@unitus.it (G.C.); 6Institute of Biomembranes, Bioenergetics and Molecular Biotechnologies, IBIOM, CNR, 70126 Bari, Italy

**Keywords:** stress, exercise, horse, gene expression, intron retention, repetitive elements

## Abstract

Physical exercise is universally recognized as stressful. Among the “sport species”, the horse is probably the most appropriate model for investigating the genomic response to stress due to the homogeneity of its genetic background. The aim of this work is to dissect the whole transcription modulation in Peripheral Blood Mononuclear Cells (PBMCs) after exercise with a time course framework focusing on unexplored regions related to introns and intergenic portions. PBMCs NGS from five 3 year old Sardinian Anglo-Arab racehorses collected at rest and after a 2000 m race was performed. Apart from differential gene expression ascertainment between the two time points the complexity of transcription for alternative transcripts was identified. Interestingly, we noted a transcription shift from the coding to the non-coding regions. We further investigated the possible causes of this phenomenon focusing on genomic repeats, using a differential expression approach and finding a strong general up-regulation of repetitive elements such as LINE. Since their modulation is also associated with the “exonization”, the recruitment of repeats that act with regulatory functions, suggesting that there might be an active regulation of this transcriptional shift. Thanks to an innovative bioinformatic approach, our study could represent a model for the transcriptomic investigation of stress.

## 1. Introduction

During the last few years, transcriptomic analysis has experienced a tremendous boost and it’s the gold standard for fully characterize the RNA molecules expressed in a certain tissue in a specific physiological time.

Great efforts and resources have been invested in defining the functional part of the genome (which turned out to be remarkable) through the extensive use of RNA-Sequencing (RNA-Seq) technology in humans [1] and in animals with other ongoing initiatives such Functional Annotation of Animal Genomes (FAANG) [2].

Despite new knowledge has been produced and genome annotations significantly improved, the genome response, in terms of gene expression modulation, is far from being understood. The big annotation projects, with some exceptions (FANTOM Projects [3]), focused mostly on “spatial” variation, intended as variation of expression in different cells and tissues, while the “temporal” one, physiological and pathological variation through time, has been left behind. This is particularly true for “non-model” species.

Also different studies (Genome Wide Association Studies—GWAS) pointed towards the intergenic and intronic fraction of the genome [4] where the most regulatory molecules lie and are transcribed.

One of the most promising approach in the dissection of the molecular basis of a phenotype is the Expression Quantitative Trait Loci (eQTL), where variations on gene expression are associated to specific regions or variants in the DNA. Studies in eQTL are already possible and are being applied, but the real step forward in functional genomics could be made by analyzing temporal data other than only the spatial ones, understanding “when” and “how much” the existing DNA variation could affect the transcriptional machinery, and therefore the phenotype. This is especially true when we are facing with biological problems involving a complex multi-organ process such exercise and stress related organism response.

Time-course experiments may allow to fully understand the functional response of the genome to stimuli when pathways or set of genes and their regulators change their action along the time: this is the case of physiological adaptation to exercise.

The horse (*Equus ferus caballus*) is *de facto* one of the few domestic species shaped by evolution to be athletic. Man has enhanced this natural predisposition through selection, leading to the modern breeds, where performance, coded in a variety of sport disciplines, is the actual phenotype of interest.

The horse, indeed, has larger lungs, a greater blood volume with a splenic reserve, a high aerobic capacity due to a high number of muscular mitochondria and large deposits of muscle glycogen with respect to other species [5]. In addition to these metabolic characteristics, the musculoskeletal structure represents an energetically efficient biomechanical system: the distal extremities of the limbs contain little muscle mass and numerous tendon structures that reduce the energy required for their movement and promote the accumulation of elastic energy [6].

These aspects, homogeneous background and natural aptitude to athletic effort, make the equine species a unique model to inspect the molecular response to stress such as heavy training and racing events [7,8,9].

Strenuous exercise stress and athletic performance are obviously strictly correlated and often, during the quest for the maximum performance, health is threatened leading to low performance syndromes such as overreaching or even overtraining [10].

In this study we concentrate our efforts on a well-established experimental design [8,11,12,13,14,15,16,17] to investigate the less studied portion of the genome that contributes, with a variety of mechanisms and/or molecules, to the exercise-induced stress response. Besides alternative splicing, intron retention [18], nonsense-mediated decay (NMD) [19] and transposons transcription [20] are often overlooked in a transcriptome study. Following previous intuition and evidences [17,21] we decided to take a look beyond the genes and describe how transcription is modulated immediately after exercise-induce stress of a gallop race in horse (Figure 1).

## 2. Materials and Methods

### 2.1. Training

All the horses recruited in this study followed the same training schedule. Before the competitions, horses were subjected to a daily training (6 days out of 7) on racetrack for about an hour. Each athlete was warmed up with 15 min of walk and trot, then the aerobic protocol was applied: progressive speeds form walk-to-canter with series lasting 1–3 min. Starting three days before the race, a strength workout that consisted of a canter-to-gallop at progressive speeds for 100–200 m (anaerobic workout) was added to previously described protocol. After competition, subjects were allowed to rest for one or two days, depending on the effort.

### 2.2. Sampling

Sardinian Anglo-Arab horses (five, two males and three females) participating in gallop races (2000 m) were recruited for the study. Ages ranged from three to four years. Peripheral blood samples were taken via jugular venipuncture at two time points: T0, at rest before the competition and at T1, immediately after the competition. Samples were taken in the same day, by one authorized veterinary, at the Pinna di Sassari racecourse (Sardinia, Italy) during the routine anti-doping procedures following the guidelines contained in the DM 797 of 16th October 2002 and subsequent amendments of Ministry of Agricultural, Food and Forestry Policies (MiPAAF). In addition, informed consent was given to horses’ owners. PBMCs were isolated through a gradient centrifugation on Ficoll-Hypaque (GE Healthcare, Pollards Wood, UK) immediately after sample collection, to avoid changes that may occur on one side and possible bias in the gene expression due to cellular heterogeneity on the other. Once obtained, cells were immediately cryopreserved (−80 °C) in TriZol until RNA extraction.

### 2.3. RNA Extraction

After cell separation, samples were subjected to RNA extraction with a Fatty and Fibrous RNA Kit (cat. 732-6870, BIO-RAD, Hercules, CA, USA) following manufacturer specifications. The extracted RNA was quantified by VersaFluor-Bio-Rad fluorimeter using the Quant-It RNA kit (Invitrogen, Dorset, UK) and quality was verified through microfluidic electrophoresis (Bioanalyzer2100 Agilent Technologies, Santa Clara, CA, USA).

### 2.4. Sequencing

Samples were sequenced using Illumina technology. Illumina TruSeq 2 libraries for a total of 10 samples, 5 T0 and 5 T1 were prepared. Massive parallel sequencing was carried out on an HiSeq 1500 machine generating 101 bases paired-end reads.

### 2.5. Bioinformatic Analyses

RAW sequences from the sequencer were checked for quality control and trimmed from adapters using FastQC (https://www.bioinformatics.babraham.ac.uk/projects/fastqc/) and Trimmomatic v. 0.33 [22], respectively. Hisat2 [23] was used for mapping reads on the reference genome equcab3 [24] deposited in National Center for Biotechnology Information (NCBI) under the Bio-Project PRJNA605934, with the following name sequence and accession number: S10B, SAMN14082184; S10G, SAMN14082185; S5B, SAMN14082186; S5G, SAMN14082187; S6B, SAMN14082188; S6G, SAMN14082189; S8B, SAMN14082190; S8G, SAMN14082191; S9B, SAMN14082192; S9G, SAMN14082193.

#### 2.5.1. Annotations Retrieval and Count Matrices

The complete Ensembl genes annotation (Release 98) was obtained through the Ensembl web page (https://www.ensembl.org/Equus_caballus/Info/Index). This file was used as starting point to construct the count matrices, one for the exons one for the introns. To assess for the exact proportion of reads aligning to intron regions we created a custom “negative” GTF file. Briefly, we constructed two preparatory GTF files through bash commands starting from the downloaded annotation: one file contained start/end coordinates for genes while the other one for each exon. Then we used bedtools subtract [25] to obtain the difference between overlapping regions. Therefore, the resulting GTF obtained by subtracting exons to genes contains only the introns, whatever the gene structure is. This procedure accounts also for overlapping genes, avoiding the user to choose which transcript is representative in presence of multiple isoforms. Some minor parsing adjustments were performed to produce a formally correct GTF. Counts were obtained with featureCounts [26] program using the computed GTF files.

#### 2.5.2. Differential Expression Analyses: Genes

The differential gene expression analysis on the two produced datasets (gene counts from exons and from introns) was carried out through the R DESeq2 package [27] that offers a method for gene-level analysis of RNA-seq data. We set as selection parameters an absolute Fold Change (log2FC) greater than 1 and an adjusted p smaller than 0.05. (q < 0.05) [28]. Before the differential analysis, data were filtered for scarcely expressed entries, excluding genes whose mean expression was less than 5 FPKM in at least 80% of samples.

Through the execution of a custom Python script, we compared the differentially expressed genes (DEGs) for both exon and intron regions with a list of genes belonging to gene ontology (GO) referred to splicing events and mRNA maturation such as: regulation of alternative mRNA splicing via spliceosome (GO:0000381); mRNA splicing via spliceosome (GO:0000398); RNA processing (GO:0006396); RNA splicing (GO:0008380); RNA splicing regulation (GO:0043484).

#### 2.5.3. Differential Expression Analyses: Isoforms

We assessed the differentially expressed gene isoforms between the two biological conditions (T0 and T1), using DEXseq [29] a Bioconductor package that allows differential exon usage estimation. This analysis was carried out both for both exon and intron compartment.

#### 2.5.4. Differential Expression Analyses: Repetitive Elements

Differential repeats analysis was carried out in three phases:(1)Genome-wide differential expression analysis(2)Differential expression analysis of repeats classes(3)Differential expression analysis of long interspersed nuclear elements subclass 1 (LINE1) only.

Full repeats representation on equcab3 were retrieved from the University of California Santa Cruz (UCSC) Table Browser [30]. Minor bash parsing procedures were applied to produce a formally correct GTF for counting purposes with featureCounts, preserving name, class and family of the repeated element. The “genome-wide” differential expression of repeats was performed as previously described for exon and intron compartments respectively. Repeats class level aggregate statistics were also produced summing the results from DESeq2 output.

The relation between differentially expressed repeats and differentially expressed genes (exon and introns) was assessed through coordinate intersection using BEDTools [25]. Functional enrichment analysis was performed on the resulting genes subsets.

For full-length LINE-1 elements characterization, the LiftOver tool by UCSC Genome Browser [31] was used to update a custom annotation (GTF file provided by Professor David Adelson, University of Adelaide, personal communication) referred to the previous genome assembly equcab2 to the new one (complete sequences are available in Appendix A). Successively, we carried out a differential expression analysis as described above for the genome-wide repeats analysis.

## 3. Results

### 3.1. Sequencing Statistics

The results of sequencing and the related quality controls are summarized in Table 1. The average sequencing depth is about ~22,000,000 reads per sample. Of these, more than 97% passed the trimming phase and more than 86.4% have a single match in the genome. Only these reads were used for downstream analyses. 

Interestingly, analyzing the reads alignment rate and comparing the number of reads assigned to exon and intron portions, we noticed a transcriptional shift from the exons to the introns in T1 with respect to T0. On average, the percentage of exons-mapping reads in T0 was 56.3% and lowered to 53.4% in T1 (paired *T*-Test, *p* = 0.03) while the same on the intron-mapping reads was 23.2% in T0 and 27.6% in T1 with an increment of 4.4% (paired *T*-Test, *p* = 0.008). Only significant results are reported. The reads mapping into the repeated elements also increased from 14.3% in T0 to 15.8% in T1 (paired *T*-Test, *p* = 0.03). The remaining portion of the aligned reads was located in intergenic regions Table 2.

### 3.2. Differential Expression Analyses: Genes

Two differential expression analyses, one for the exon and one for the intron regions were carried out, setting T0 as baseline. After statistical analysis with DESeq2 software, starting from a cleaned dataset of 15,038 genes, 648 were found as differentially expressed in exonic compartment (with q < 0.05), 396 of which were up-regulated (log2FC ≥ +1) and 252 were down-regulated (log2FC ≤ −1), full results are provided in Figure 2, Table 3 and Appendix A. For the intronic compartment, 1306 were differentially expressed, 637 up-regulated and 669 down-regulated in T1 vs. T0 (Figure 3, Table 3 and Appendix A). Comparing the common list of DEGs from exons and introns (1697 entries) with the list of GO terms referring to splicing events (673 entries), 44 modulated genes involved in messenger RNA maturation processes emerged. Of these, 10 are within the exonic compartment, 30 within the intronic ones and 4 presented a “full gene” expression modulation (Appendix A).

### 3.3. Differential Expression Analyses: Isoforms

From the isoform evaluation using DEXSeq, 150 genes emerged that had at least one differentially expressed exon in the sample after the race compared with those at rest (Appendix A). It is intriguing that 139 of these genes were not differentially expressed in the previous analysis (exon and intron compartment).

We investigated the biological functions of this gene subset with a GO enrichment analysis using String DB (https://string-db.org) that showed 160 interactions between 129 proteins retrieved from the 139 host genes (Appendix A). The analysis pointed out 20 statistically significant enriched biological processes (q < 0.05). Among them, some were in line with our biological framework: “cellular response to stimulus”, “response to stress”. Other processes were implicated in the gene expression regulation, immune system activation and cardiac and striatal muscles regulation.

Using DEXSeq software to examine the intron differential expression we found 2997 introns differentially expressed contained in a total of 2020 genes (Appendix A). Although most of them were poorly expressed, this approach pointed out that a substantial part of the differentially transcribed regions is usually overlooked.

### 3.4. Differential Expression Analyses: Repetitive Elements

We also evaluated the differential expression of repetitive elements, starting from a cleaned dataset comprising 66,151 repeats. In total, 7439 elements were significantly differentially expressed (q < 0.05) with an absolute log2FC greater or equal than 1 [4939 UP-regulated and 2500 DOWN-regulated] (Figure 4, Appendix A).

From the evaluation of the repeat class we noticed that, almost half of the modulated repeats belong to the LINE (46.7%), followed by short interspersed nuclear elements (SINEs) (23.8%), DNA repetitive elements (11.7%) and long terminal repeats (LTR) (9.1%). Only a small amount of these repetitive elements is located within intergenic portion, while the remaining part (over 80%) is localized within gene sequences. The 94.1% of these (that correspond to the 77.3% of the total number of differentially expressed repeats) are within intron portion of the genes; the repeat class ranking remained the same of the genome wide distribution with a small increment of LINE percentage (Appendix A). Interestingly, the analysis highlighted that over half (60%) of these genes is not modulated in T1.

The intersection between differentially expressed repeats and differentially expressed introns showed 153 matches that are retained intron contained at least a modulated repeat within its sequence.

These introns belong to 143 genes corresponding and GO enrichment analysis of this group of genes exhibited several biological enriched processes such as “response to stress” and “response to stimulus”, involving 45 and 74 genes, respectively (Appendix A).

The full-length LINEs analysis allowed us to identify 19 entire LINE1 elements modulated in T1 samples (Appendix A).

## 4. Discussion

The genomic response induced by exercise is carried out through different mechanisms where transcription modulation is the king. With our experimental design we tried to dissect the transcription modulation in horse PBMCs after exercise focusing also on unexplored regions related to introns and intergenic portions.

In humans, as in other species, transcriptional changes induced by stress events have been extensively studied and, although they mainly concern protein-coding genes transcription modulation, many other effects have been noticed. For example the induction of transcriptional read-through that generates very long downstream sequences of gene-containing transcripts (DoGs) [32] or the production of non-coding RNAs [33] and changes in chromatin structure [34]. Under stress conditions, post transcriptional mRNA processing can also be altered [35].

Another phenomenon called exonization or intron retention, that is the transcription of introns through alternative splicing also induced by the transcription of local transposed elements, can be activated [36,37]. Repetitive elements are indeed known to influence the surrounding sequences once activated [38]. All these mechanisms represent a sort of genomic response to external stimuli, and we tried to search for signs of them in exercise induced stress.

Our first result highlights the difference in transcripts mapping to exonic o intronic gene portions: we noticed a statistically significant transcriptional shift after a race (T1), with a reduction in the exonic reads and an increase in the intronic sequences in comparison with T0 (Figure 1).

Considering the number and the modulation of genes focusing exonic part (Figure 2) 648 were differentially expressed in T1 compared to T0.

If we focus on intron sequences, usually spliced and less functional, and we found over 1306 genes differentially expressed as part of the intronic component (Figure 3). According to this result, we tried to evaluate if this was the consequence of alternative splicing activation (AS). AS indeed is the major source of protein diversity and appear to be the general rule rather than an exception; it is now well known that alternative splicing is widely used by the cell [1] and, in case of homeostasis perturbation, the cell can apply a cost-effective strategy preferring to enhance an advantageous isoform, or a not functional one, instead of modulating or shutting down the entire transcription machinery [19]. AS is a highly regulated process with a complex interactions network between sequence features within the pre-mRNA [39] and trans-active splicing factor proteins [40]. This process can occur physiologically for steady state cell protein production, but also in case of response to developmental cues and external stimuli, including stress [41,42].

To further investigate if the transcriptional shift in our data was due to AS towards not annotated exons or to co-transcriptional splicing [43], we matched our DEGs with those having GO terms referred to splicing: 44 modulated genes involved in mRNA maturation processes emerged, encouraging us to continue the investigation aimed at alternative splicing and its potential causes (Appendix A).

Indeed, the isoform analysis allowed us to identify the transcription of each exon. From this analysis, 150 genes emerged, reduced then to 139 considering only those that did not appear in the 648 DEGs from the exon compartment (Appendix A).

With these genes, that are preferentially expressing a particular isoform in the post-race, we performed a GO enrichment analysis to assess which biological processes were mostly involved. Several categories related to response of exercise-induced stress emerged: *response to stimulus, cellular response to stress, positive regulation of gene expression, regulation of muscle adaptation* and many others listed in Appendix A. This result can be interpreted as a confirmation that exercise-induced stress activates targeted alternative splicing in genes belonging to key biological processes.

The same analysis was carried out for the detection of isoforms on the intronic compartment to evaluate the stress-induced intron transcription that may be correlated to the intron retention (IR) phenomenon, considered one of the multiple declinations of AS [44]. With IR a normally excluded intron is maintained in the final mRNA transcript, probably due to weak splice signals [45,46]. The result of this mechanism is a post-transcriptionally gene-expression regulation through the NMD resulting in the open reading frame interruption by the introduction of premature termination codons [19]. In other cases, IR has the classical alternative splicing function that is to generate protein diversity allowing the production of pairs of protein isoforms [18].

We found 2020 genes with differentially expressed introns therefore with putative retained intron (Appendix A). This result confirms that our data are largely interested by this phenomenon.

In literature a general increase of intron retention events has been observed in response to some types of stress like thermal stress exposure [47,48], chemotherapy-induced stress in cancer cells [49] and hypoxia [50]. The intron retention phenomenon could be triggered by the transcriptional activation of repeated elements contained within the intron sequence [36,37]. These features represent over 40% of the human genome sequence [51], 46% of equine ones [52], and are widespread both in genic and intergenic portions. LINEs, long interspersed nucleotide elements, are among the most represented Transposable Elements (TEs) in the genome, making up about 20% of human DNA sequence [51] (17% in the horse [52]). TEs are so called because of the ability to be integrated into other parts of the genome via an RNA intermediate.

For their capabilities to transcribe and induce intron retention, we investigated the repeated sequences with a differential analysis approach that allowed us to identify over 7000 modulated repeats, half of which were LINE 1 (Figure 4). Interestingly, four fifths of these modulated repeats were within gene sequences, and within differentially expressed genes, confirming the not random nature of their expression (Figure 1). Moreover, 153 differentially expressed introns had at least one differentially expressed repeat within them. These results may support the hypothesis that intron retention is facilitated by the containing of repeated elements, whose transcriptional increase probably activates the exonization phenomenon.

Only a small part of LINE sequences is able, as well as transcribing, to actively retrotranspose, indeed most of them are in a truncated form in the genome.

Moreover, retrotransposition does not normally occur in somatic cells because the repeats are strongly silenced by a high degree of methylation [53,54], but in high-stress conditions the methylation level can decrease and the LINE1 transcription can influence the intron retention [20,55,56,57,58].

LINEs subclass L1 transcription is driven by an internal RNA pol II promoter [59] which allows it to copy itself in an RNA intermediate pasting it into another genome location. It is mandatory, however, that the L1 sequence is intact and functional to replicate itself [60].

While contributing to genomic variability and species evolution this behavior may induce several adverse events [20,55,56,57]. Since these genomic modifications may be harmful, the host cell is provided with several mechanisms to suppress dangerous retrotransposon activity [53,54], among which one of the main known is the methylation of CpG sequences in the LINE-1 5′ UTR. In fact, a general LINE-1 methylation decrease has been observed in case of intense stress conditions such as many cancer types [58] and neurological disorders and, indeed, the retrotransposition is demonstrated only in few tissues (germinal and nervous tissues) [38,61].

For these reasons, it may be interesting to evaluate the transcription of the entire LINE1 that have the ability, once completely transcribed, to copy and paste itself in other parts of the genome. In our data there were 19 LINE1 (Appendix A) elements entirely transcribed and therefore potentially retrontranscribed open the scenery of retrotransposition caused by exercise-induced stress. Retrotransposition indeed can lead to gene breaks and this could explain what referred to as derangement of the immune system during the overtraining syndrome [62].

These observations may be at the base of the genomic plasticity in response to stress or in particular stress stimulus such as that induced by strenuous physical effort.

## 5. Conclusions

Our results strongly suggest that strenuous exercise leads to stress adaptation through a variety of genomic mechanisms from the activation of typical stress response pathways and genes (inflammatory and immune response) to the activation of additional resources that stimulate genome plasticity. These include targeted alternative splicing activation, increased intron transcription through intron retention, repeats transcription, especially within genes, and LINE1 full-length modulation that can lead to retrotransposition as ultimately resource of genomic plasticity under stress conditions.

## Figures and Tables

**Figure 1 genes-11-00410-f001:**
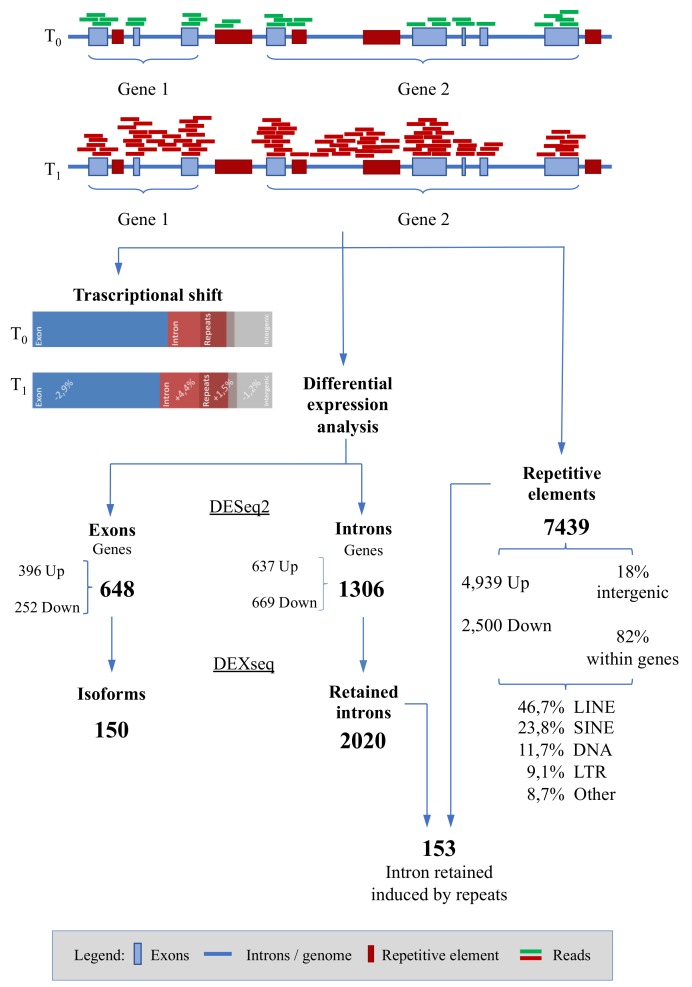
Cartoon depicting results in the context of the method workflow.

**Figure 2 genes-11-00410-f002:**
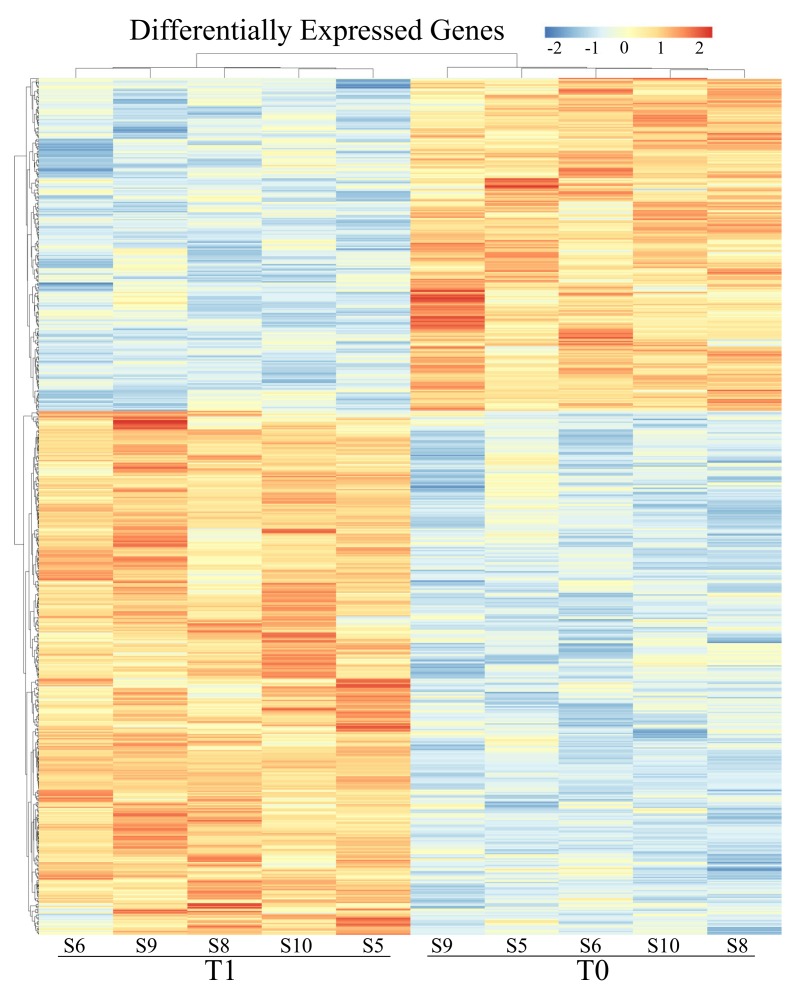
Heat maps of differentially expressed genes basing on exon counts. Clustering resulted to be accurate and distinct. Legend above displays range of expression.

**Figure 3 genes-11-00410-f003:**
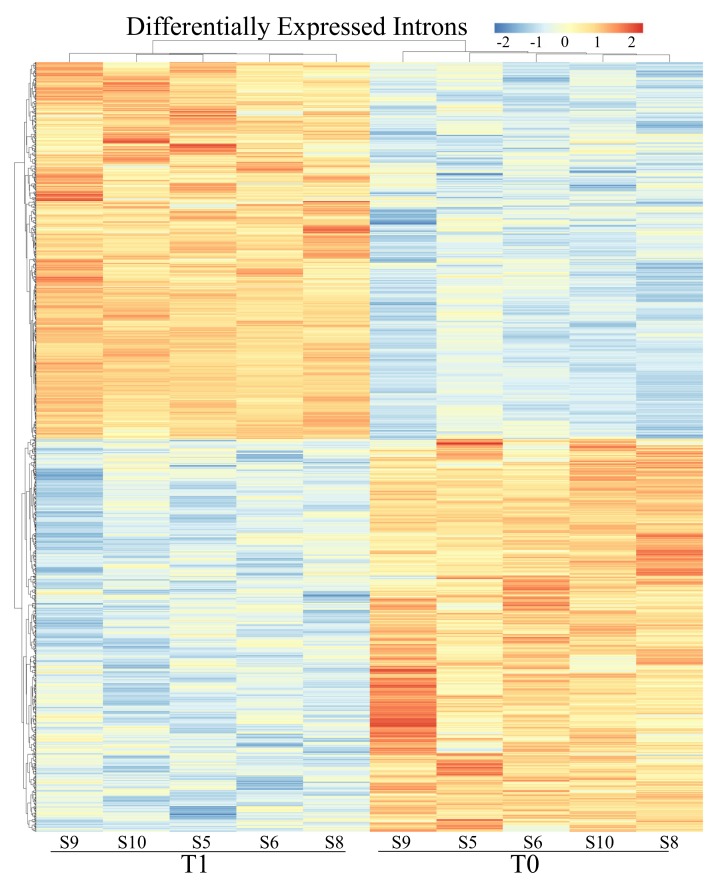
Heat maps of differentially expressed genes basing on intron counts. Like for the DEGs for the exonic compartment, also in this case there is a clear sample clustering in the two biological conditions. Legend above displays range of expression.

**Figure 4 genes-11-00410-f004:**
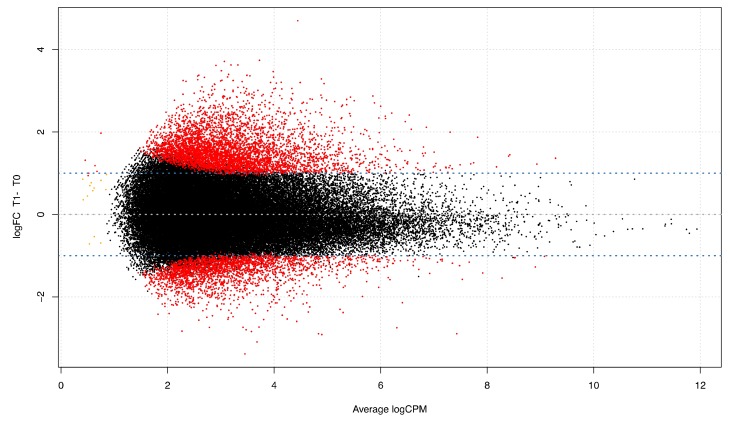
Smear plot of differentially expressed repetitive elements genome wide, red dots indicate the entries that overcome the fold change threshold (Y axes) and are significantly (q < 0.5) different between T0 and T1. X axes represent the expression value within the library.

**Table 1 genes-11-00410-t001:** Summary of sequencing results.

Sample	Reads before Trimming	Reads after Trimming	Quality Check Passed Rate (%)	Uniquely Mapped Reads	Alignment Rate (%)
S5_T0	18,986,960	18,391,294	96.9	7,930,946	86.2
S5_T1	19,209,286	18,644,550	97.1	8,112,987	87.0
S6_T0	19,143,702	18,542,082	96.9	7,833,923	84.4
S6_T1	15,681,066	15,223,494	97.1	6,619,839	87.0
S8_T0	20,252,456	19,654,560	97.0	8,542,631	86.9
S8_T1	31,767,674	30,848,206	97.1	13,349,560	86.6
S9_T0	15,724,850	15,241,064	96.9	6,598,277	86.6
S9_T1	25,053,756	24,340,766	97.2	10,589,195	86.7
S10_T0	20,330,330	19,746,518	97.1	8,631,228	87.4
S10_T1	33,503,070	32,465,956	96.9	14,217,209	87.4
Average	21,965,315	21,309,849	97.0	9,242,580	86.60

**Table 2 genes-11-00410-t002:** Reads alignment on genomic compartments.

Sample	Total Alignments	Successfully Assigned Alignments EXONS	%	Successfully Assigned Alignments INTRONS	%	Successfully Assigned Alignments REPEATS	%
S10_T0	9,418,426	5,220,406	55.4	2,282,112	24.2	1,397,171	14.8
S5_T0	8,731,754	4,697,224	53.8	2,233,626	25.6	1,342,363	15.4
S6_T0	8,600,396	4,899,846	57	1,863,096	21.7	1,145,141	13.3
S8_T0	9,313,192	5,146,610	55.3	2,268,959	24.4	1,423,386	15.3
S9_T0	7,236,941	4,332,077	59.9	1,442,060	19.9	921,749	12.7
Average	8,660,141.80	4,859,232.60	56.28	2,017,970.60	23.16	1,245,962.00	14.30
S10_T1	15,479,241	8,015,665	51.8	4,616,059	29.8	2,603,246	16.8
S5_T1	8,858,376	4,813,955	54.3	2,389,440	27	1,377,956	15.6
S6_T1	7,230,807	3,883,746	53.7	1,976,927	27.3	1,138,745	15.7
S8_T1	14,650,606	7,835,416	53.5	3,893,566	26.6	2,256,959	15.4
S9_T1	11,642,249	6,259,752	53.8	3,167,804	27.2	1,806,643	15.5
Average	11,572,255.80	6,161,706.80	53.42	3,208,759.20	27.58	1,836,709.80	15.80
Race VS. Basal			−2.86		+4.42		+1.50
*t*-Test			0.029		0.008		0.028

**Table 3 genes-11-00410-t003:** Top 20 up and down differentially expressed genes and introns.

	Genes		Introns	
	ID	log2Fold Change	ID	log2Fold Change
Upregulated	ENSECAG00000030595	10.53	ENSECAG00000018841	6.29
	ENSECAG00000019352	7.58	ENSECAG00000039315	6.09
	ENSECAG00000001516	6.98	ENSECAG00000020003	5.39
	ENSECAG00000038063	6.24	ENSECAG00000017073	4.79
	ENSECAG00000023163	6.24	ENSECAG00000011929	4.49
	ENSECAG00000009129	5.97	ENSECAG00000004515	4.33
	ENSECAG00000009755	5.96	ENSECAG00000002619	4.30
	ENSECAG00000021383	5.75	ENSECAG00000010669	4.00
	ENSECAG00000011929	5.57	ENSECAG00000005905	3.95
	ENSECAG00000020402	5.54	ENSECAG00000030110	3.91
	ENSECAG00000034297	5.48	ENSECAG00000003573	3.86
	ENSECAG00000039315	5.45	ENSECAG00000010860	3.77
	ENSECAG00000033016	5.34	ENSECAG00000016321	3.68
	ENSECAG00000015766	4.67	ENSECAG00000013594	3.66
	ENSECAG00000020003	4.53	ENSECAG00000000051	3.66
	ENSECAG00000040244	4.51	ENSECAG00000039959	3.57
	ENSECAG00000002234	4.36	ENSECAG00000014979	3.56
	ENSECAG00000010860	4.31	ENSECAG00000023173	3.47
	ENSECAG00000033856	4.31	ENSECAG00000009215	3.43
	ENSECAG00000015992	4.00	ENSECAG00000015318	3.37
Downregulated	ENSECAG00000032106	−23.04	ENSECAG00000040402	−4.91
	ENSECAG00000034632	−6.08	ENSECAG00000023475	−4.33
	ENSECAG00000007460	−5.64	ENSECAG00000028489	−4.14
	ENSECAG00000021087	−4.55	ENSECAG00000011895	−3.99
	ENSECAG00000010281	−4.17	ENSECAG00000031371	−3.62
	ENSECAG00000009895	−4.07	ENSECAG00000032756	−3.58
	ENSECAG00000035315	−3.54	ENSECAG00000036032	−3.56
	ENSECAG00000008274	−3.40	ENSECAG00000036704	−3.48
	ENSECAG00000009869	−3.22	ENSECAG00000000386	−3.45
	ENSECAG00000032544	−3.09	ENSECAG00000017676	−3.43
	ENSECAG00000032503	−3.04	ENSECAG00000025038	−3.34
	ENSECAG00000030934	−2.90	ENSECAG00000022510	−3.34
	ENSECAG00000034974	−2.85	ENSECAG00000006114	−3.28
	ENSECAG00000009625	−2.78	ENSECAG00000027794	−3.26
	ENSECAG00000034925	−2.72	ENSECAG00000011660	−3.18
	ENSECAG00000010326	−2.68	ENSECAG00000014338	−3.12
	ENSECAG00000036608	−2.63	ENSECAG00000033795	−3.10
	ENSECAG00000037661	−2.55	ENSECAG00000033519	−2.99
	ENSECAG00000014338	−2.53	ENSECAG00000011723	−2.97
	ENSECAG00000015083	−2.48	ENSECAG00000036290	−2.85

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
