# Peer review of "Gallop Racing Shifts Mature mRNA towards Introns: Does Exercise-Induced Stress Enhance Genome Plasticity?"

_genes, 2020, doi:10.3390/genes11040410_

Round 1
Reviewer 1 Report
Cappelli et al. have performed two time-points RNA-seq of peripheral blood mononuclear cells, before and after a competition horserace, to study effects of extreme exercies on transcription and – via transcription of LINE elements – potentially on genome stability.
After a standard initial processing of the RNA-seq data they performed an innovative later steps of analysis, aimed at elucidation of global effects of athletic exercise on differential expression not only of exons but also of introns. This analysis permitted conclusion that exercie leads to increased relative transcription of introns and global activation of alternative splicing, in particular exonisation or intron-retention.
Importantly, the authors intersected their dataset of differentially expressed introns with the genomic locations of repetitive DNA elements, in particular transposable elements (TE), identifying 153 retined introns that contained differentially expressed TE within their sequence. Among the set of differentially expressed TE the authors identified 19 full-length transcripts of LINE elements, opening a possibility that some of them are retrotranscriptionally competent – i.e. that athletic exercise may locally in the genome activate retrotransposition.
Overall, this is an elegant investigation, although limited by its correlation-only nature. Some aspects of the analysis and the manuscript should be improved before it is suitable for publication.
Analysis:
1. The significance of changes in the percenteges of exonic and intronic read alingnments before (T0) and after exercie (T1) could also be compared using paired T-test, with pairing by sample. Only the results from the version of the T-test leading to lower p-values need to be reported.
2. Percenages of particular classes of TE among the differentially expressed repetitive elements (section 3.4) should be compared not only with their genome-wide content, as provided in the Discussion, but also with their intronic content – to see if there is over-representation of certain classes of TE among their differentially expressed subset.
3. Assembly version(s) corresponding to genomic coordinates in Table S9 should be given.
4. Check if the 19 full-length LINE transcripts contain non-mutated LINE-specific ORFs,
or provide as supplementary information the sequences of the reads mapped to these LINE elements enabling others to perform this check.
5. (Optional) A relationship between intron retention and presence of differentially expressed TE described in lines 224/226 could be illustrated in detail for an example gene locus, by showing the stacked aligned paired-reads before and after the race.
6. Terminology
The term “genome plasticity” (lines 3-4, 42, 267) should preferably only be used the narrower sense of structural change of the genome (DNA), already well established for procaryots and also in TE literature (e.g. PMID: 15148416). A more general term e.g. “genomic response” (as used in line 253) should be used in lines 42 and 267. Without this distinction, the key observation of this work, namely that the athletic exercise not only affects transcription, but – by transcriptionally activating full-length LINE elements – potentially also affects the sequence of the genome, cannot be sufficiently precisely stated.
Minor grammatical/lexical issues:
There are in several places grammatical/lexical problems (possibly of the reviewer – but double check), which hamper understanding of some of the key sentences:
line 70: “attitude to athletic effort” – possibly “aptitude to athletic effort”
line 81: “fog of war”– possibly drop or explain
line 175: “22,000,000 million” – double check and possibly correct
line 224-226 – This sentence is hard to understand. Possibly, there should be no coma before “that” (a defining relative clause) and “at list” is a typo for “at least”?
Line 263: “Another phenomenon called exonisation or intron retention” should be changed to “Other phenomena called ...” as they two phenomena are not the same thing.
Reviewer 2 Report
Gallop race shifts mature mRNA towards introns: does exercise-induced stress enhance genome plasticity
This study looks at the change in the transcriptional landscape of peripheral blood mononuclear cells within Sardinian Arab-Anglo racehorses before and after intensive exercise. The authors key findings where identifying transcriptional shift from coding to non-coding regions and further categorising the proportion of these events that occur within repetitive elements.
Overall this is a well thought out study. The methodology is sound and the findings are important for the broader field of understanding transcriptional response following stress. I am happy to recommend this article for publication and have no major comments. I do have a handful of minor comments which I believe will help with the overall clarity of the manuscript.
- Figure1 and Figure2 – Heatmaps could be more clearly labelled showing T0 and T1. Currently these plots describe (i) that the data separates well between T0 and T1, and (ii) that there are two consistent clusters of up and down-regulated genes or introns. I think the authors could expand on these figures to further explain the biological changes. Including an insert showing the top 20 up/down differentially expressed genes/introns.
- Ontology analysis is described in the text but the top hits could be summarised as a bar chart so the details for odds ratio, confidence and significance are more easily accessible.
- Figure 4 – I found this to be a very useful figure to visually summarise the change in gene expression pre and post exercise as well as the core findings. As a first time reader, I felt this would have been more useful at the beginning of the manuscript rather than at the end.
- The authors have made their FASTQ files available on NCBI. In the spirit of sharing I encourage the authors also make available their custom scripts/pipelines used to process this data.
